# Comparative Mitogenomics and Phylogenetic Implications for Nine Species of the Subfamily Meconematinae (Orthoptera: Tettigoniidae)

**DOI:** 10.3390/insects15060413

**Published:** 2024-06-03

**Authors:** Siyu Pang, Qianwen Zhang, Lili Liang, Yanting Qin, Shan Li, Xun Bian

**Affiliations:** 1Key Laboratory of Ecology of Rare and Endangered Species and Environmental Protection, Guangxi Normal University, Ministry of Education, Guilin 541006, China; pangsiyu08200@163.com (S.P.); zhangqianwen@163.com (Q.Z.); lll188145@163.com (L.L.); qinyanting2019@163.com (Y.Q.); shanli694@163.com (S.L.); 2College of Life Sciences, Guangxi Normal University, Guilin 541006, China

**Keywords:** Orthoptera, Meconematinae, mitochondrial genome, phylogenetic analysis

## Abstract

**Simple Summary:**

The mitochondrial genome (mitogenome) has been used extensively to investigate phylogenetic relationships at various taxonomic levels. Insect mitochondrial genomes are important for understanding insect evolution and relationships. We performed complete mitogenome sequencing and annotation for nine Meconematinae species and combined the data with data from previously assembled mitochondrial genomes in the family Tettigoniidae to conduct a phylogenetic analysis. The monophyly of *Xizicus*, *Xiphidiopsis*, and *Phlugiolopsis* was not well supported in our findings.

**Abstract:**

Currently, the subfamily Meconematinae encompasses 1029 species, but whole-mitochondrial-genome assemblies have only been made available for 13. In this study, the whole mitochondrial genomes (mitogenomes) of nine additional species in the subfamily Meconematinae were sequenced. The size ranged from 15,627 bp to 17,461 bp, indicating double-stranded circular structures. The length of the control region was the main cause of the difference in mitochondrial genome length among the nine species. All the mitogenomes including 13 protein-coding genes (PCGs), 22 transfer RNA genes (tRNAs), 2 ribosomal RNA genes (rRNAs) and a control region (CR). The majority strand encoded 23 genes, and the minority strand encoded 14 genes. A phylogenetic analysis reaffirmed the monophyletic status of each subfamily, but the monophysitism of *Xizicus*, *Xiphidiopsis* and *Phlugiolopsis* was not supported.

## 1. Introduction

Meconematinae is a speciose subfamily of the Tettigoniidae which is mainly found in Asia, Australia, and the Pacific Islands, and is a diverse group of organisms that also contain the smallest katydids that live in tropical forests [1]. This subfamily was originally described by Burmeister in 1838 based on the genus *Meconema* Serville, 1831 [2], which is a small genus that primarily inhabits tropical and subtropical regions and has not received significant attention [3]. The controversy over tribe or genus classification of Meconematinae has never ceased [4,5]. However, the number of newly discovered Meconematinae genera and species continues to increase [6,7,8,9,10].

Studies of DNA barcoding in invertebrates have shown that using only DNA barcoding data can be misleading regarding biodiversity when sampling is inadequate and classifying is disputed [11,12]. In addition, there are uncertainties in the study of the evolutionary process of speciation [13]. The cases in which a small number of metazoan animals are genetically polylineal or polylineal cannot be distinguished by barcoding [14]. The mitochondrial genome contains not only the information of DNA barcoding, but also other, more phylogenetic signals, which makes up for the shortcomings of DNA barcoding. The advantages of small mitochondrial genomes, rapid rates of evolution, low sequence recombination, and conserved gene products have been widely used as molecular markers for phylogenetic analysis and evolutionary genome research [15,16,17,18,19]. Although mitochondrial genomes can deeply solve phylogenies, their usefulness as markers for highly divergent lineages is still controversial [20]. Insect mitochondrial genomes are circular, usually 14 kb to 18 kb in length, and encode 37 genes: 13 protein-coding genes (PCGs), 2 rRNAs, and 22 tRNAs. In addition to the 37 genes, there is one control region (CR), which is the starting point of replication and transcription and is also referred to as the A + T rich region in the literature [18]. Due to the high mutation rate of mitochondrial DNA, it can be a very useful molecule for high-resolution analysis of the evolutionary processes of species [21]. Meanwhile, the insertion of pseudogenes (that is, non-functional fragments) in mitochondrial genes in Orthoptera is less likely, but they contain more effective information [22].

Research has found that *Xizicus* in the Meconematinae was the earliest differentiated clade in Tettigoniidae [23]. Meanwhile, the Meconematinae is a sister to all other Tettigoniidae [23], but contrary to the results of other studies, the Nedubini may be a sister to all Tettigoniidae [24]. Despite the large number of species in Meconematinae, the number of assembled mitogenomes documented for species of this subfamily is still limited, and only ten species belonging to seven genera have had their mitogenomes sequenced. Although the monophyly of Meconematinae has been established, there is still some controversy over the relationships among the genera within this subfamily [3,15,25,26,27,28,29,30]. Phylogenetic analysis by Mao et al. (2020) revealed that (((*Xizicus* + *Xiphidiopsis*) + (*Pseudokuzicus* + *Shoveliteratura*)) + (*Pseudocosmetura* + *Acosmetura*) + *Decma*) [3], while phylogenetic conclusions made by Han et al. (2019), such as (((*Pseudocosmetura* + *Acosmetura*) + (*Xizicus* + *Xiphidiopsis*)) + *Pseudokuzicus*) + *Decma*) [30], revealed that the relationships among different genera were not well resolved. The monophyly of *Xizicus* has not been confirmed [3,30]. Previous studies have indicated that the phylogenetic positions of the genus *Decma* in maximum likelihood (ML) and Bayesian inference (BI) trees are inconsistent [3,15]. Due to the limited mitochondrial genome data available for Meconematinae, the inconsistent results have not satisfied the need for phylogenetic analyses of different genera. In this study, high-throughput sequencing technology was used to sequence and annotate the genomes of nine species in the Meconematinae. A phylogenetic analysis of the mitochondrial genome data of Tettigoniidae downloaded from NCBI database was performed to reveal the relationships among the genera of Meconematinae.

## 2. Materials and Methods

### 2.1. Specimen Extraction and Sequencing

In this study, total DNA was extracted from the hind femur muscles of nine species of Meconematinae according to the TIANamp Genomic DNA Kit. The specimens extracted this time were collected by the research team in the field and stored at Guangxi Normal University. DNA samples were sent to Berry Genomics (Beijing, China) for Illumina NovaSeq high-throughput sequencing.

### 2.2. Mitochondrial Genome Sequence Screening and Assembly

Nine new mitochondrial genomes were assembled via the CLC Genomics Workbench [31] by comparing the high-throughput sequencing data with the whole-mitochondrial-genome sequences of closely related species in the NCBI database [32]. The sequences with the highest homology with the sequencing data were selected as the reference genes and then assembled using NOVOPlasty4.1.2 [33].

### 2.3. Annotation and Analysis of the Mitochondrial Genome

MITOS2 of the Galaxy tool online service was used to identify tRNAs and predict secondary structures [34]. Then, manual correction was performed according to the mitochondrial genomes of related species. Circular maps of the mitochondrial genome were created and constructed online via the CGView-Circular Genome Viewer website [35]. To predict tandem repeats in the control region, the Tandem Repeat Finder website was utilized [36]. The total length of the mitochondrial genome, 13 PCGs, 2 rRNAs, 22 tRNAs, and CR were calculated using MEGA11.0 software for base composition and base skew rate [37]. The ratio of the number of synonymous substitutions per synonymous site (Ks) and the number of nonsynonymous substitutions per nonsynonymous site (Ka) of 13 PCGs was computed using DnaSP5 software [38]. PhyloSuite v1.2.3 was used to generate four datasets [39]: (I) PCG123: PCGs with all three codon positions; (II) PCG12: PCGs with the 1st and 2nd codon positions; (III) PCG123R: PCG123 dataset plus two rRNAs; and (IV) PCG12R: PCG12 dataset plus two rRNAs. Heterogeneity and substitution saturation were assessed for each dataset using AliGROOVE v1.08 and DNAMBE v5 [40,41].

### 2.4. Construction of Phylogenetic Trees

The phylogenetic analysis were performed combining the nine newly sequenced mitogenomes and 88 species of Tettigoniidae downloaded from GenBank. The outgroup taxa were *Sosibia gibba* and *Sosibia ovata*.

MAFFT in PhyloSuite v1.2.3 software was used for auto multiple-sequence alignment [39,42], and then the concatenation function of PhyloSuite v.1.2.3 was utilized to generate concatenated files for the genes. ModelFinder was used to refer to the Akaike information standard to perform the optimal partitioning strategy and model selection for the concatenated data [43]. The “mrbayes” and “all” were selected for the model parameters, and “linked branclengths” was selected to reduce the time. MrBayes was used for the BI tree, and IQ-TREE was used for the ML tree [44,45], with both tools using the GTR model [46]. Bayesian inference analysis utilizing MrBayes was applied for phylogenetic reconstruction with the following settings: 2,000,000 generations, four numbers of chains, trees sampled every 1000 generations, and an initial 25% of trees discarded as burn-in [44]. Phylogenetic analyses were conducted with IQ-TREE, employing the maximum likelihood (ML) method with automatic model selection and 1000 standard bootstrap replicates for tree support [47]. The resulting phylogenetic tree was visually edited using the online website iTOL and is presented as a phylogenetic tree diagram [48].

## 3. Results and Discussion

### 3.1. Comparison of Mitochondrial Genome Lengths

The nine newly assembled mitochondrial genome sequences were all circular structures, as shown in Appendix A. The size of the genomes ranged from 15,627 bp to 17,461 bp (Figure 1), and the gene arrangement was the same as that of the inferred ancestral insect mitochondrial genome [13]. The differences in the lengths of the mitochondrial genomes among the nine species were mainly attributed to differences in the lengths of the control regions. The relative position and transcription direction of each gene were consistent.

### 3.2. Mitochondrial Gene Interval and Overlapping Regions

The mitochondrial genes had interval and overlapping regions. There were 8 intergenic spacers in *Chandozhinskia hastaticercus*; 9 in *Paraphlugiolopsis jiangi*, *Phlugiolopsis punctata*, *Phlugiolopsis brevis*, and *Phlugiolopsis tuberculata*; 10 in *Phlugiolopsis tribranchis*, *Grigoriora cheni*, and *Xizicus fascipes*; and 11 in *Microconema* sp. The intergenic spacers of the nine mitogenomes ranged from 1 bp to 21 bp in size; the longest located between trnS2-nad1 in *Grigoriora cheni* was 21 bp and that in the other species was 16 bp. The length of the *trnS2*-*nad1* intergenic spacer in *Chandozhinskia hastaticercus* was 18 bp, while in the other 8 species, it was 17 bp. The trnP-nad6 intergenic spacers in all the species were 1 bp in length.

Long intergenic noncoding spacers (100–500 bp) have also been found in Lepidoptera [49], Coleoptera [50], and other orthopterans [15,19]. The number of overlapping regions in the nine species was greater than the number of interval regions. There were 12 overlapping regions in *Microconema* sp.; 13 in *Phlugiolopsis punctata*, *Phlugiolopsis brevis*, *Xizicus fascipes,* and *Phlugiolopsis tuberculata*; 14 in *Chandozhinskia hastaticercus* and *Paraphlugiolopsis jiangi*; 15 in *Phlugiolopsis tribranchis*; and 17 in *Grigoriora cheni*. There were fewer overlapping bases in the overlapping regions than in the spacer regions. The number of overlapping bases in 11 overlapping regions in the nine species was consistent; that is, there was an overlap of 1 bp between *trnK*-*trnD*, *atp6*-*cox3*, *trnA*-*trnR*, *trnT*-*trnP,* and *nad6*-*cob*; *nad2*-*trnW* and *cob*-*trnS2* overlapped 2 bp; *trnI*-*trnQ* overlapped 3 bp; *atp8*-*atp6* overlapped 7 bp; and both *trnW*-*trnC* and *trnY*-*cox1* overlapped 8 bp.

### 3.3. Nucleotide Composition and Skew

The AT content and skew statistics are shown in Appendix A. The base content of the mitochondrial genomes from the nine species showed a significant A + T bias, with the content ranging from 68.99% to 73.61%. The lowest content was found in *Phlugiolopsis tribranchis*, and the highest content was found in *Grigoriora cheni*. The strong A + T-bias might have stemmed from external environmental factors [51]. The A + T content in PCGs (69.06–72.84%), tRNAs (74.12–75.15%), rRNAs (73.51–75.32%), and CRs (60.5–74.69%) was greater than the G + C content, indicating a bias towards the A + T content in the nucleotide composition of the nine species (Appendix A). Additionally, the AT-skew had a positive value, and the GC-skew had a negative value for the complete mitochondrial genome, which implied that A was more abundant than T and that C was more abundant than G (Appendix A).

PCGs usually use the typical initiator codon ATN, and the special initiator codon TTG occurred only as the start codon of *nad1* in seven of the species, with *Phlugiolopsis tribranchis* and *Xizicus fascipes* being the exceptions. The complete terminator codon TAA and the incomplete terminator codons T and TA often occurred, and the complete terminator codon TAG occurred only for *cob* and *nad1*. The *nad4l* start codon was ATG, and the stop codon was TAA. The *cob* start codon was ATG, and the end codon was TAG. Notably, an incomplete codon functions after post-transcription polyadenylation and is converted into a full-stop codon [52]. The longest gene, *nad5*, was 1732 bp in length, except in the case of *Chandozhinskia hastaticercus*, in which the length was 1733 bp. Some genes were constant, such as *nad2* at 1029 bp, *atp6* at 678 bp, *nad3* at 354 bp, *nad4l* at 297 bp, *nad6* at 528 bp, and *cob* at 1137 bp.

### 3.4. Protein-Coding Genes and Codon Usage

All the mitogenomes contained the typical gene content, including 13 PCGs, 22 tRNAs, 2 rRNAs, and a CR. Twenty-three genes were encoded on the majority strand, while the remaining fourteen genes were encoded on the minority strand. Twenty-two tRNAs were responsible for transporting 20 amino acids, and both serine and leucine were transported by two tRNAs. The tRNAs ranged in size from 63 bp to 71 bp. Two rRNAs (*rrnL* and *rrnS)* encoded 16S rRNA and 12S rRNA, respectively. The relative position and transcription direction of each gene were consistent, and no gene rearrangement was detected.

### 3.5. tRNA and rRNA Genes

The use of the tRNA anticodon was the same for all nine species. Twenty-two tRNAs ranged in size from 63 bp to 71 bp, and the lengths of some tRNAs were constant. For example, the lengths of *trnW*, *trnK*, *trnS1*, *trnS2,* and *trnV* were 66 bp, 70 bp, 61 bp, 69 bp, and 71 bp, respectively (Appendix A). In addition to *trnI, trnL2*, *trnN*, *trnS1*, *trnE,* and *trnS2*, G–U (or U–G) pairs were found in the secondary structures of other tRNAs. There were 21 cases in *Grigoriora cheni* and *Chandozhinskia hastaticercus*, 24 cases in *Xizicus fascipes*, 25 cases in *Phlugiolopsis brevis* and *Phlugiolopsis tuberculata*, 26 cases in *Microconema* sp., 28 cases in *Phlugiolopsis punctata*, 30 cases in *Paraphlugiolopsis jiangi*, and 31 cases in *Phlugiolopsis tribranchis*. There was a G-U pair on the anticodon arm of *trnM*, *trnC*, *trnY*, *trnV*, *trnP*, *trnG*, *trnH*, *trnF,* and *trnA* for all nine species. The secondary structure of *trnS1* in *Phlugiolopsis tribranchis*, *Paraphlugiolopsis jiangi*, *Phlugiolopsis brevis,* and *Phlugiolopsis tuberculata* lacked a D arm. The secondary structure of *trnS1* lacked a D loop and D arm in *Grigoriora cheni*, *Chandozhinskia hastaticercus*, *Microconema* sp., *Phlugiolopsis punctata,* and *Xizicus fascipes* (Appendix A).

rRNAs were encoded by minority strands. Notably, *rrnL* was located between *trnL1* and *trnV*, and *rrnS* was located between *trnV* and the CR. The base length of *rrnL* ranged from 1308 bp to 1316 bp, and that of *rrnS* ranged from 787 bp to 792 bp.

### 3.6. Control Region

Tandem duplication is when a piece of DNA is converted into two or more copies, each of which follows the preceding one in a contiguous fashion [36]. There were tandem repeats in the CRs of nine species. The repetition length and number of each species were different, among which *Grigoriora cheni* had the longest repetition length of 230 bp, and *Phlugiolopsis tribranchis* had the shortest repetition length of 13 bp (Figure 2). There was one tandem repeat in the CR of *Phlugiolopsis brevis* and *Xizicus fascipes*. Two tandem repeats in the CR were found in *Phlugiolopsis tuberculata*. Three were found in *Microconema* sp., *Paraphlugiolopsis jiangi*, *Chandozhinskia hastaticercus*, and *Phlugiolopsis punctata*. Four were found in *Grigoriora cheni*. There were five tandem repeats in the CR of *Phlugiolopsis tribranchis*. Each sequence of repeats is under the influence of local and general biological activities that determine its level of instability [53]. Repeating sequences are common in most metazoans, and it has also been observed that the mitochondrial DNA of closely related species does not share the same repeating sequence [54]. Therefore, the analysis of repeated sequences between different individuals can reveal population structure information regarding the species and provide clues to the phylogenetic relationships [55].

### 3.7. The Rate of Evolution of 13 PCGs

Non-synonymous substitutions can give rise to defects in respiratory-chain activity that reduce the efficiency of metabolic processes and are generally harmful [56]. The harmful effect of mitochondrial non-synonymous mutations triggers highly effective purifying selection, which maintains the fitness of the mitogenome [57]. The ratio of the number of synonymous substitutions per synonymous site (Ks) and the number of nonsynonymous substitutions per nonsynonymous site (Ka) for 13 PCGs from the nine species were calculated to estimate the evolutionary rate of the PCGs (Figure 3). The results were all less than 1, indicating that purifying selection among the 13 PCGs played a major role. The Ka/Ks value of *cox1* was the lowest. This indicates that *cox1* was subject to lower purification selection and was not susceptible to environmental impact. It was the most conserved and the slowest to evolve among the 13 PCGs, followed by *cob* and *cox3*. The Ka/Ks value of *nad6* was the highest, and it was susceptible to environmental impact, so the evolutionary rate of *nad6* was the fastest among the 13 PCGs, followed by those of *atp8* and *nad4*.

Locomotion is energy-consuming, and strongly locomotive organisms usually require a more active metabolism than weakly locomotive organisms. Previous studies have shown that those without flying ability accumulate more non-synonymous substitutions and have higher Ka/Ks values than those with flying ability, and flightless species experienced more relaxed evolutionary constraints [58]. Among the twenty species in the Meconematinae subfamily (Appendix A), the Ka/Ks value of *atp8* in the macropterous region (from wing to abdomen or longer than the abdomen) and brachypterous were the highest, with the fastest evolution rate (Appendix A). The Ka/Ks values of the macropterous and brachypterous *cox1* genes were the lowest, followed by *cox2* and *cytb* (Appendix A). Moreover, the Ka/Ks value of *cox2* in *Pseudokuzicus pieli,* a macropterous species, was the highest (Appendix A). The average values of the two groups were all <1, which showed that the mitogenomic evolution of different wing types was not neutral and was subjected to the pressure of purifying selection. The Ka/Ks values of the brachypterous type were lower than those of the macropterous type, and flightless insects accumulated fewer nonsynonymous mutations (Appendix A). Contrary to the study by Chang et al., flightless orthoptera insects accumulated more non-synonymous mutations than flying orthoptera insects [59].

### 3.8. Substitution Saturation Tests and Nucleotide Heterogeneity

The index of substitution saturation (Iss) for the four nucleotide sequence datasets of 99 species was significantly lower than that of the critical Iss (Iss.cSym and Iss.cAym values in symmetrical and asymmetrical trees, respectively). It showed that the third codons of 13 PCGs were not saturated and contained phylogenetic information (Appendix A). Heterogeneity in nucleotide divergence was evaluated via pairwise comparisons in multiple sequence alignment. There was little heterogeneity among the four datasets (Appendix A).

Several studies have shown that the third codon position can be removed because it is highly saturated and has fewer informative sites [60,61], whereas other studies have suggested that the third codon position is phylogenetically informative and thus should be considered in phylogenetic studies [62,63]. Our findings indicate that the third codon position did not affect the topology of the phylogenetic tree. The results of BI tree topology based on PCG123 and PCG12 were basically the same, and the BI tree based on PCG123 data set had high support among all subfamilies. Thus, the role of the third codon position requires consideration in phylogenetic studies and should not be excluded on the assumption that it is not informative, as has been the case in many previous studies.

### 3.9. Phylogenetic Analysis of Meconematinae

Based on PCG123 and PCG123R in the mitochondrial genomes of 99 species, including 9 newly described species, BI and ML phylogenetic trees were reconstructed. These provided strong support for the monophyletic relationships between subfamilies, but the relationships between each subfamily varied greatly. The BI tree (PPs: 1-0.849) and the ML tree, which were based on PCG123R (BPs: 100): (Meconematinae + (Tettigoniinae + Bradyporinae)) + Conocephalinae) + (Listroscelidinae + Lipotactinae)) (Figure 4, Appendix A), are consistent with the results of previous studies [3]. The ML tree was constructed based on PCG123 (BPs: 46): (Meconematinae + (Listroscelidinae + (Conocephalinae + Lipotactinae)) + (Tettigoniinae + Bradyporinae)) (Figure 5). Previous studies have shown that clades with a BS of 50–69 or BPP of 0.85–0.89 are considered weakly supported, and clades with a BS of <50 or BPP of <0.85 are considered unsupported [64]; therefore, the sister relationships of the ML tree based on PCG123 are not tenable. *Xiphidiopsis gurneyi* was shown to be closely related to *Xizicus maculatus*, which is consistent with the findings of previous research [5,21]. *Xizicus howardi* did not cluster with other species of *Xizicus*, *Xiphidiopsis,* and *Paraphlugiolopsis* when placed in the *Xizicus* and *Phlugiolopsis* complexes and failed to form a clade. *Paraphlugiolopsis* was erected according to the apices of posttibiae with two pairs of spines [65], while the monophyly of the genus was not well supported in our findings. The phylogenetic tree, which is based on PCG123, showed that *Decma* was the first to evolve, which is consistent with the findings of previous research [3,25,26,27,30].

## 4. Conclusions

The complete mitochondrial genomes of nine Meconematinae species were assembled and annotated in this study. The Ka/Ks values of the brachypterous type were lower than those of the macropterous type, and flightless insects accumulated fewer nonsynonymous mutations. This research provides some reference for the study of different wing types of Orthoptera. All four topological analyses confirmed that each subfamily was monophyletic. However, the monophyly of *Xizicus*, *Xiphidiopsis,* and *Phlugiolopsis* was not supported. This study will help to advance the study of Meconematinae phylogeny.

## Figures and Tables

**Figure 1 insects-15-00413-f001:**
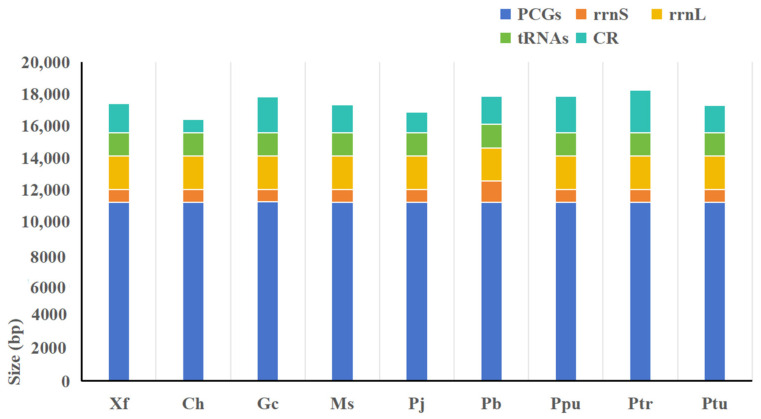
Size comparison of protein-coding genes (PCGs), transfer RNA genes (tRNAs), rrnL, rrnS, and control region (CR) among 9 Meconematinae mitogenomes. Species are abbreviated as follows: Xf: *Xizicus fascipes*; Ch: *Chandozhinskia hastaticercus*; Gc: *Grigoriora cheni*; Ms: *Microconema* sp.; Pj: *Paraphlugiolopsis jiangi*; Pb: *Phlugiolopsis brevis*; Ppu: *Phlugiolopsis punctata*; Ptr: *Phlugiolopsis tribranchis*; Ptu: *Phlugiolopsis tuberculata*.

**Figure 2 insects-15-00413-f002:**
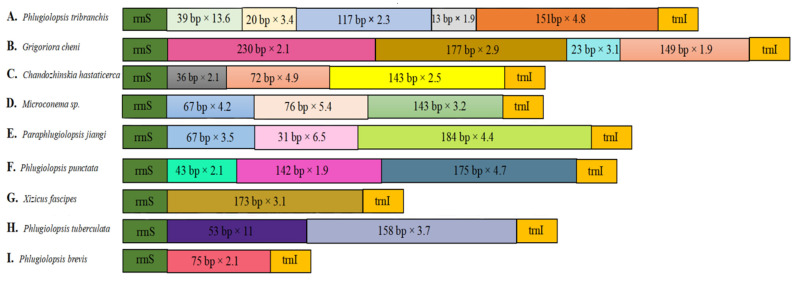
Schematic diagram of tandem repeat arrangements in the CR of nine species of the subfamily Meconematinae (**A**–**I**). Species names are annotated on the left side of the figure. Different colors in the box represent different tandem repeat sequences, and the length and number of repetitions are indicated in the box.

**Figure 3 insects-15-00413-f003:**
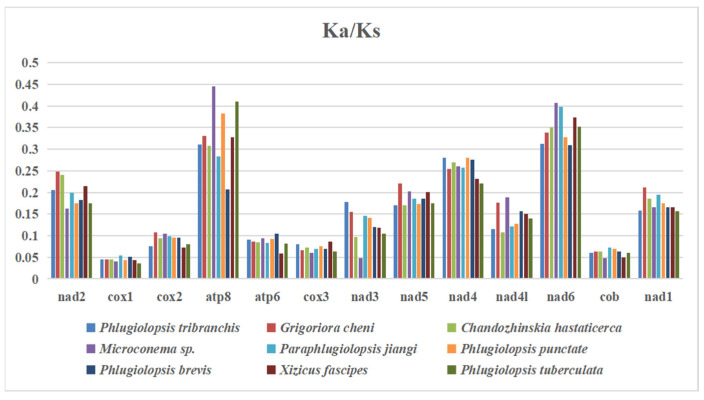
Ka/Ks values of 13 PCGs in nine species of Meconematinae. The bar indicates each gene’s Ka/Ks value.

**Figure 4 insects-15-00413-f004:**
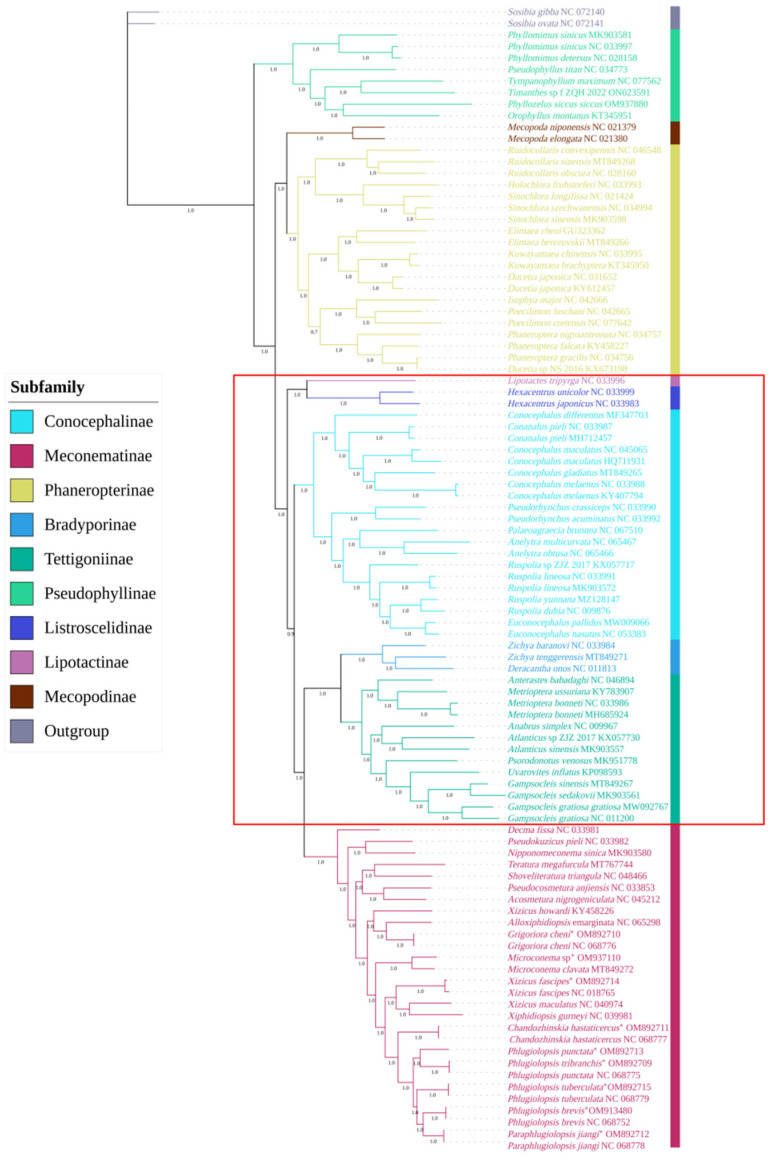
Phylogenetic tree was obtained based on BI analysis of 13 PCGs from 99 Tettigoniidae mitochondrial genomes. The relationships between the subfamilies in the red square were inconsistent with the results in Figure 5.

**Figure 5 insects-15-00413-f005:**
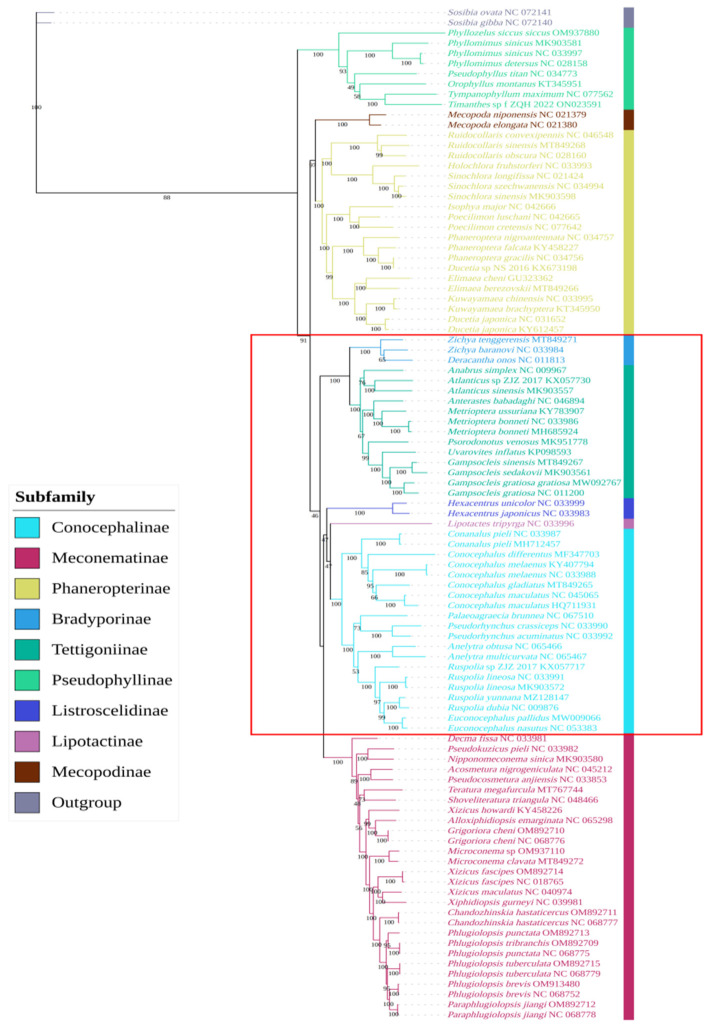
Phylogenetic tree obtained based on ML analysis of 13 PCGs from 99 Tettigoniidae mitochondrial genomes. The relationships between the subfamilies in the red square were inconsistent with the results in Figure 4.

## Data Availability

The newly sequenced nine mitogenome sequences have been submitted at NCBI (Acc. number OM892709, OM892710, OM892711, OM937110, OM892712, OM8913, OM892714, OM892715, OM913480).

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
