# Peer review of "Comparative Mitogenomics and Phylogenetic Implications for Nine Species of the Subfamily Meconematinae (Orthoptera: Tettigoniidae)"

_insects, 2024, doi:10.3390/insects15060413_

Round 1

Reviewer 1 Report

Comments and Suggestions for Authors

In their analysis Pang et al. present novel mitochondrial genomic data for nine species of the orthopteran subfamily Meconematinae and together with available data reassess that phylogenetic relationship of this group. The methods used seem adequate and the added genomic data is an important boost for providing essential genomic data for a greater share of orthopteran. I note that I have rather an ecological background so most of my comments address the implications of the findings of this study and their ecological relevance. In that sense I believe the paper is not sufficiently well-written to address a less specialized readership. This does not only imply less clarity but also less generality. Please take particular care with structuring and rewording the results & discussions part. In fact, I highly recommend splitting this section into a results section, including the detailed and technical results and a short more generally appealing discussion. An important aspect to take in mind is really that the reader should be able to understand all the concepts and measure from this text alone, which in your case requires, substantial rewording and introductory sentences before the finding and the respective discussion. These authors find this and those found that is rhetorically poor and does not help to understand your main message/finding. In addition, I recommend including given aspect of the ecology/distribution or similar that can be used to grasp the importance of more high-resolution genomic data. If you run an analysis of the latitudinal range center of species with just COI data /or a recently published phylogeny and compare it to your more resolved one, for instance, the value of your work become clearer.

Please find detailed comments in the attached pdf. For a revision this would need to be listed and addressed with line by line comments.

Kind regards. I am looking forward to reviewing a modified version of this article.

Reviewer 2 Report

Comments and Suggestions for Authors

General comments

The paper by Pang et al. describes 9 new mitogenomes for meconematine katydids. The authors justify this study well by showing the paucity of genetics known for this large subfamily of katydids. The methods seem straightforward, but I recommend expanded explanation of some methods (see comments below) so that readers may better understand the output. There are some evolutionary implications of the comparative mitogenomics found here that are potentially interesting, and I would like to see an expanded Introduction that covers topics such as the relationship between wing length, dispersal and/or gene flow, and evolutionary rate.

Introduction

Pg. 1 begins with the statement that Meconematinae hold an important place in the katydid phylogeny. Why is the phylogenetic position of Meconematinae more important, or above the importance of the position of any other katydid subfamily? In Song et al. 2016 this subfamily is sister to all other katydids, but this is not the result of some other studies: Mugleston et al. 2013 do not show Meconematinae in a basal position. Cole and Chiang 2016 show that Nedubini may be sister to all katydids. Introduce this conflict to add importance to this work.

Pg. 2 the authors predict that mtDNA will be useful for evolutionary studies because of a high mutation rate. This claim is not a new prediction. The literature shows that mtDNA has indeed been used for huge numbers of evolutionary studies largely for that reason and also for ease of obtaining high copy number mtDNA data. The paper cited also established this trend in 1979, and mtDNA was made accessible to entomologist in large part due to Simon et al. 1994.

In the introduction of mtDNA genomics in pg. 2, the authors explain the utility of this kind of genetic data. mtDNA genomics is certainly a useful and convenient tool for phylogenetic, population, and evolutionary studies. Perhaps the authors might also mention some limitaitons, for example, all mtDNA genes are linked (inherited as a single unit) and show only maternal lineage. Specific to Orthoptera, an advantage of mtDNA genomics that the authors may point out is the avoidance of mt pseudogenes, which are very common in the huge genomes of Orthoptera (see e.g. Song 2014).

Please introduce some of the concepts behind findings in the Results & Discussion. The macropterous and micropterous taxa comparison is an interesting theme that I think deserves introduction, such as with respect to gene flow and migration. It is known that geographic population structure is reflected in mtDNA, particularly for geographically isolated populations. Are taxa fixed as to wing morphology? Or do some populations contain both micropterous and macropterous individuals?

Methods

2.1 Please explain the source and reasoning behind the species and specimens used in this study. A table of GenBank accessions with specimen voucher data is missing from this work. Where were they collected? Why were 9 sequenced in this study? Were the specimens collected opportunistically or were they targeted to be phylogenetically informative. Where are the vouchers deposited?

Sequencing methods need more explanation. What high throughput sequencing platform was used? Was mtDNA captured bioinformatically from whole genome sequencing? What was the intended coverage? Was low coverage genome skimming employed? Skimming is problematic given the large genomes of Orthoptera, where organelle genomes are a much smaller percentage of the DNA extract.

2.4 Please explain the outgroup choice of Sosibia. Do other phylogenies show this as a basal taxon?

The authors report using model testing and partitioning approaches. The Methods then state that the GTR model was used for all analyses. If GTR was chosen by model tests, this is a Result and belongs in the Results/Discussion section. It is not surprising that GTR was returned as the model with a complex multigene dataset.

Results & Discussion

Please add sequencing QC statistics, such as from QUAST. Minimum results may consist of the coverage, N50, or other statistics that show the reader the quality and quantity of the output.

Consider reporting model testing/partitioning output. How many partitions were returned by ModelFinder? Was the choice of GTR justified by the AIC criterion?

3.2 and 3.3 both describe intergenic spacers. This information is better if grouped together in a single pg.

3.5. First sentence says tRNA codon usage was the same. The same as what? Of each other? To other katydid mtgenomes that have been studied?

3.7. This section contrasts macropterous and micropterous species, and uses the term “region” for those classifications based on wing length. Macropterous and micropterous are morphs or forms, not regions.

I like the contrast of these morphs, gene flow is affected by mobility that would account for the higher rate of evolution found for micropterous taxa in this study.

Specific comments

Simple summary

Line 6. “Tettigoniidea” Was this taxon name intended? Elsewhere Tettigoniidae is used.

Abstract

L7 “the mitogenomes contained typical, including...” what was typical? Presumably the expected gene content.

Comments on the Quality of English Language

Introduction

Pg. 1 L6-7 consider moving comma for clarity, changing to “Serville, 1831 [4], which is a small genus that primarily inhabit tropiocal and subtropical regions, is not highly relevant to humans, and has not received significant attention [5].

Round 2

Reviewer 1 Report

Comments and Suggestions for Authors

Dear Authors,

Thank you for considering my previous comments and suggestions. I believe the manuscript has significantly improved. However, I feel that the first paragraph of the introduction still needs some refinement. Currently, it begins very specifically with details on various topological hypotheses for the group of interest. I highly recommend using an indirect citation style instead of phrases like 'XY et al. found that...' and generalizing your statements more.

For example, you could begin with a general statement about the impact of improving genetic data, and then state that data quality and availability have profoundly influenced the taxonomy of your group of interest, mentioning the type of data they used mophology, barcodes, etc. If possible also add some statements on specific characteristics that might have led to these changes.

You might find this study on birds helpful in illustrating how improved genetic resolution can significantly affect estimates of evolutionary rates, such as the phylogenetic signals of traits (https://www.nature.com/articles/s41586-024-07323-1; Fig. 5a).

Congratulations on this fine paper! If these changes are taken seriously, I believe they can be left to the discretion of the authors.

Comments on the Quality of English Language

The quality of English language is sufficient.

Author Response

I am very sorry that my mistake has brought some trouble to your work.
Thank you very much for your comments and suggestions on my article, and I have revised it accordingly. Please see the attachment.
